# Etelcalcetide in Patients on Hemodialysis with Severe Secondary Hyperparathyroidism. Multicenter Study in “Real Life”

**DOI:** 10.3390/jcm8071066

**Published:** 2019-07-20

**Authors:** Domenico Russo, Rocco Tripepi, Fabio Malberti, Biagio Di Iorio, Bernadette Scognamiglio, Luca Di Lullo, Immacolata Gaia Paduano, Giovanni Luigi Tripepi, Vincenzo Antonio Panuccio

**Affiliations:** 1Department of Public Health, University of Naples FEDERICO II, 80131 Naples, Italy; 2Institute of Clinical Physiology (IFC-CNR) Research Unit of Reggio Calabria, 89124 Reggio Calabria, Italy; 3Department of Nephrology Cremona Hospital, 26100 Cremona, Italy; 4Department of Nephrology AORN Cardarelli, 80131 Naples, Italy; 5Department of Nephrology Ospedale “Parodi Delfino” di Colleferro (Roma), 00034 Colleferro, Roma, Italy; 6Nephrology, Dialysis and transplantation Unit G.O.M. “Bianchi Melacrino Morelli”, 89121 Reggio Calabria, Italy

**Keywords:** secondary hyperparathyroidism, cinacalcet, etelcalcetide, hypocalcemia, gastrointestinal side effects

## Abstract

Etelcalcetide is a new calcimimetic indicated for the treatment of secondary hyperparathyroidism (SHPT) in dialysis patients. Etelcalcetide efficacy in SHPT has been ascertained only in randomized controlled trials. This multicenter study was carried out in “real world” setting that is different from randomized controlled trials (RCTs) to (1) evaluate the effectiveness of etelcalcetide in SHPT, (2) to assess calcium, phosphorus, alkaline phosphatase changes, (3) to register gastrointestinal side effects. Data were collected from twenty-three dialysis units with *n* = 1190 patients on the charge. From this cohort, *n* = 168 (14%) patients were on treatment with etelcalcetide, and they were evaluated for statistics. A median weekly dose of etelcalcetide was 15 mg (7.5–45 mg). Patients were either naïve (33%) or switched from cinacalcet to obtain better control of SHPT with reduced side effects or pills burden. Serum parathyroid hormone (PTH) declined over time from a median value of 636 pg/mL to 357 pg/mL. The median time for responders (intact PTH (iPTH) range: two to nine times the upper normal limit) was 53 days; the percentage of responders increased (from baseline 27% to 63%) being similar in switched-patients and naïve-patients. Few patients had symptomatic hypocalcemia requiring etelcalcetide withdrawal (four cases (3%) at 30-day control, two cases (2%) at 60-day, one case (1%) at 90-day control). Side effects with etelcalcetide were lower (3–4%) than that registered during cinacalcet treatment (53%). Etelcalcetide is a new therapeutic option for SHPT with low side effects and pills burden. Etelcalcetide may improve adherence to therapy, avoiding unremitting SHP. It remains to be assessed whether etelcalcetide may reduce parathyroidectomy, vascular calcification, or mortality. Being etelcalcetide very potent in suppressing PTH levels, even in severe SHPT, future studies should evaluate the potential risk of more adynamic bone disease during long-term therapy.

## 1. Introduction

Secondary hyperparathyroidism (SHPT) is frequent in patients on dialysis.

Pathogenesis of SHPT is multifactorial; therefore, several therapeutic strategies have to be utilized to reduce serum levels of parathyroid hormone (PTH), such as calcitriol, oral and IV vitamin D analogs, phosphate binders, and calcimimetic cinacalcet.

For many years, cinacalcet, the unique calcimimetic available, has been regarded as the preferred option for the subgroup of patients with SHPT characterized by hypercalcemia, or with SHPT refractory to calcitriol or active vitamin D analogs.

Recent guidelines do not prioritize any medication, but they suggest calcimimetic, calcitriol, or vitamin D analogs, or a combination of calcimimetic with calcitriol or vitamin D analogs in patients on dialysis with SHPT [1]. Therefore, no restrictions exist for the use of calcimimetic in treating SHPT.

Etelcalcetide is a new calcimimetic that can be given intravenously. The efficacy of etelcalcetide in reducing levels of PTH has been ascertained in randomized controlled trials (RCTs) [2,3]. No data are available on the effectiveness and safety of etelcalcetide in everyday clinical practice.

This multicenter study evaluates as primary endpoint the effectiveness of etelcalcetide in reducing levels of PTH in a “real world” setting that is different from RCTs. Furthermore, the study assesses as secondary endpoints the changes in serum levels of calcium, phosphorus, alkaline phosphatase, and the occurrence of gastrointestinal side effects during the treatment with etelcalcetide.

## 2. Study Design

This observational study was conducted in patients from hospital and university dialysis clinics, and private dialysis units located in different geographic areas of our Country.

Patients were 18-years-old or older; they were on stable doses of active vitamin D analogs, phosphorus binders, a supplement of oral calcium, and calcium concentration in dialysate (1.25–1.50 mmol/L).

Patients with long QT syndrome, in pregnancy or nursing, or who had undergone parathyroidectomy were not treated with etelcalcetide.

Etelcalcetide was administered to naïve patients (i.e., patients never treated before with calcimimetic) with SHPT and to patients who were on treatment with cinacalcet; in the latter condition, etelcalcetide was administered as replacement medication (switch-group) either to obtain a better control of serum levels of PTH and to reduce side effects or pills burden.

On the basis of their judgment and data analysis, clinicians modified the dose of etelcalcetide (5 mg three times per week, as suggested by manufacturing company); in addition, according to summary of product characteristics and allowed co-interventions in RCTs, clinicians had taken into consideration the possibility to adjust the dosage of phosphate binders, vitamin D sterols, supplement of calcium, and calcium concentration in dialysate if needed.

The interval between assessments was established by each clinician.

Symptomatic hypocalcemia, gastrointestinal disorders, or any other side effects imputed to etelcalcetide were reported.

The following levels of serum calcium were used for the definition of hypocalcemia: <7.0 mEq/L; ≥7.0 but ≤7.5 mEq/L; ≥7.5 but <8.3 mEq/L. These levels of serum calcium are in accord with the scale recently adopted by Bushinsky et al. [4].

Clinicians were asked on a gold range of PTH levels they attempted to obtain in clinical practice.

Blood samples were collected on the post-weekend day. In almost all units, biochemical data were repeated every other week during the first two months of treatment with etelcalcetide. For the present statistical analysis, monthly data were taken into consideration.

Biochemical data were collected before the initiation of therapy with etelcalcetide; these data were regarded as baseline values for statistical analysis. Timing for following the assessment of biochemical data was established by clinicians based on their judgment.

Biochemistry was measured at local laboratories.

Assay procedures did not change throughout the reported observation periods. For PTH, measured by bio-intact assay (single dialysis unit), the correction was made [5].

Presence of abnormalities of the QT interval was verified by electrocardiograms.

### Statistical Analysis

For descriptive purposes, data are summarized as mean ± standard deviation (SD), median and interquartile range (IQR), or as percent frequency, as appropriate.

*P* for the trend of intact PTH (iPTH), serum calcium, serum phosphate, alkaline phosphate over time was obtained using Linear Regression Models weighted for patients’ identification. In detail, to assess whether iPTH, serum calcium, serum phosphate, and alkaline phosphate significantly change over time, we fitted, for each biomarker and considering the whole spectrum of available data, a linear regression model weighted for the patient’s identification to control for repeated measurements and having the above mentioned risk factors as dependent variables and time as independent variable. The *P*-value of the time variable in each model was considered as an index of the statistical significance of the temporal trend of each biomarker.

Data analysis was performed by SPSS for Windows Version 13 (IBM, Chicago, IL, USA).

## 3. Results

Data were collected from twenty-three dialysis units with *n* = 1190 patients on the charge. From this cohort, *n* = 168 (14%) patients were on treatment with etelcalcetide, and they were evaluated for statistics.

Clinical characteristics and biochemical data of patients are reported in Table 1.

Notably, there was a high presence of patients with cardiovascular comorbidities (Table 1).

One-hundred and twelve patients (67%) received etelcalcetide as replacement of cinacalcet.

Reasons for switching patients to etelcalcetide was the hope to achieve better serum levels of PTH, reduction of side-effects, or improvement of patient’s compliance by reducing pills burden.

Before the switch to etelcalcetide, there was the following scenario: 34 patients were on treatment with cinacalcet, but they complained about gastrointestinal side effects; 22 patients had discontinued cinacalcet because of gastrointestinal side effects; four patients were poorly adherent to therapy. The 52 patients on treatment with cinacalcet were without gastrointestinal side effects, but in 33 patients (63%), there was a sub-optimal control of SHPT (iPTH >500 pg/mL) because patients tolerated a sub-optimal dose of the drug. The remaining 19 patients (37%) were switched to etelcalcetide to reduce pill burden.

The median daily dose of cinacalcet was 30 mg (30–60 mg). In the whole cohort, the median weekly dose of etelcalcetide was 15 mg (7.5–45 mg), and it remained stable during the follow-up period. 

The changes in serum concentration of iPTH are reported in Figure 1a.

Serum PTH declined over time from a median value of 636 pg/mL to 357 pg/mL. The median time for responders was 53 days (Figure 2), and the percentage of responders progressively increased during the further course of treatment (panel a). This was similar either in patients switched from cinacalcet or in naïve patients (panel b).

Clinicians participating in this study declared to take into account the latest kidney disease improving global outcomes (K-DIGO) guidelines on levels of Ca, P, PTH [1]. However, when asked on personal position, they answered to feel more confident with PTH concentration between 150–300 pg/mL. The percentage of patients with iPTH within the range of 150–300 pg/mL progressively increased during etelcalcetide therapy (from 27% to 63%).

Changes in serum calcium concentration are reported in Figure 1 (panel b).

The number (percentage) of patients with hypocalcemia during etelcalcetide treatment is reported in Table 2.

Importantly only few patients had symptomatic hypocalcemia that required withdrawal of etelcalcetide during the follow-up, as reported in Table 3.

Other causes of withdrawal are reported in the above Table 3.

The majority of patients (75%) received calcitriol during treatment with either cinacalcet or etelcalcetide; therapy with native vitamin D was reported in 5% of patients (7.5% of patients on Vitamin D treatment). The percentage of patients requiring vitamin D sterols during etelcalcetide had small and not relevant fluctuations during all observation periods. The concentration of calcium in the dialysate was not modified by clinicians during etelcalcetide treatment.

Phosphorus and alkaline phosphatase progressively declined during the treatment with etelcalcetide, as reported in Figure 1c,d.

Patients on treatment with phosphate binders (either as a single drug or as an association) before etelcalcetide were 96%; about 25% of patients were on treatment with binders containing calcium. These percentages remained unmodified during treatment with etelcalcetide.

The occurrence of gastrointestinal side effects during etelcalcetide is reported in Table 4.

The occurrence of gastrointestinal side effects with etelcalcetide was lower (3–4%) than that registered during cinacalcet treatment. Indeed, 61 patients (53%) in treatment with cinacalcet complained about gastrointestinal side effects that resulted responsible for discontinuation of treatment in 24 cases (21%) and poor compliance in three cases (3%).

Assay procedures for biochemistry and dialysis modality remained unchanged throughout the observation periods.

## 4. Discussion

SHPT has multifactorial pathogenesis; therefore, diverse therapeutic strategies are mandatory to reduce levels of PTH. Nonetheless, in clinical practice, it is difficult to achieve satisfactory PTH levels in a large proportion of dialysis patients, limiting the risks of elevated serum calcium and phosphorus [5,6,7,8].

Etelcalcetide is a new long-acting calcimimetic that can be administered intravenously in patients on dialysis three times per week at the end of the hemodialysis session.

In large RCTs, etelcalcetide resulted superior to placebo and not inferior to cinacalcet, either in reducing PTH levels or in causing side effects [2,3,4,5,9].

The RCT is an ideal design for demonstrating causality between the use of specific medicine and beneficial effects or drawbacks under well definite conditions. Nonetheless, even one perfectly designed RCT does not protect against a bias for selection, which can occur both in the way that individuals are accepted or rejected for participation in a trial, and in the way that the interventions are assigned to individuals once they have been accepted into a trial [10,11,12]. Other potential biases are delivery of interventions, measurement of outcomes, choice of problem to study or type of research to use, and sample size determination [10,11,12]. Therefore, data reported by RCTs may result, to some extent, remote from the real clinical world, and they may not be completely transposed in everyday practice. In the latter setting, clinicians prescribe the drug to all patients who may benefit from it without stringent pre-specified inclusion/exclusion criteria, rigid dosage to administer, multiple endpoints to achieve, ascertainment of superiority/non-inferiority versus a comparator or placebo, and predetermined “closing-time”.

There is an increasing recognition of the value of real-world data since they are more appropriately informative on the effectiveness, while data from RCTs are informative on the efficacy of a new study medication [10,11,12]; therefore, real-world data may be regarded as a complementary source to RCTs being attained under conditions of everyday clinical practice [10].

The present multicenter observational study shows that etelcalcetide significantly reduces levels of PTH of patients on dialysis in the setting of everyday clinical practice.

It is worth noticing that the present cohort of patients was representative of a heterogeneous population with baseline levels of PTH not so different from those of patients enrolled in RCTs and with relevant cardiovascular comorbidities [2,3,4,5,9].

The reduction of PTH levels was evident after a few weeks from etelcalcetide initiation; furthermore, levels of PTH progressively declined during treatment with an increasing percentage of patients with PTH on target.

As expected, etelcalcetide reduced the level of serum calcium. Even though hypocalcemia fluctuated during the observation periods irrespective of the level of chosen serum calcium, only in few cases, clinicians considered it appropriate to withdraw etelcalcetide.

Assuming as hypocalcemia the concentration of serum calcium lower than 8.3 mg/dL, as reported in summary of product characteristics and by other investigators [2,3,4], in the present study, the percentage of patients with hypocalcemia was lower than that reported either when etelcalcetide was compared to cinacalcet (68%) or placebo (61%) in RCTs. In contrast to RCTs, no supplementation of calcium, increments of vitamin D sterols, or increase of calcium in dialysate were done in our cohort. Furthermore, assuming the concentration of serum calcium equal to or higher than 7.9 mg/dL as the value associated to lowest mortality risk [5], the majority (>80%) of our etelcalcetide treated patients were well above this limit.

We can reasonably exclude interferences of blood collection timing (i.e., on the post-weekend day) on the observed changes in PTH and calcium concentration. Indeed, in the COSMOS cohort, there were no differences in levels of parathyroid hormone and serum calcium between mid-week and post-weekend day blood withdrawal [13].

It is worth to note the relevant difference in the occurrence of gastrointestinal side effects comparing the available RCTs and the present study. Indeed, in the setting of real life, the occurrence of side effects with etelcalcetide therapy was far lower than that reported in RCTs (10%–12%) [2,3,4]. In available comparative RCTs, the occurrence of gastrointestinal disorders was reported to be similar for etelcalcetide and cinacalcet [2,3,4]. This finding was not confirmed in our patients. The percentage of occurrence of gastrointestinal disorders was 53% in cinacalcet treated patients and 3–4% in patients treated with etelcalcetide; also, etelcalcetide was withdrawn in only one patient compared to 24 cases (21%) of cinacalcet discontinuation due to gastrointestinal disorders.

These findings may allow to suppose that etelcalcetide per se may cause fewer side effects and that it may be better tolerated compared to cinacalcet. Indeed, both medications were administered in a similar clinical setting of everyday clinical practice, in the same patients and by the same clinicians. Further explanation for low incidence of side effects may be that the absence of rigid pre-specified dose to administer by protocol (for instance, investigators were blinded to PTH levels in RCTs), and absence of stringent targets to timely achieve by protocol could have allowed our clinicians to change dosage without mandatory intervals and more appropriately titrate etelcalcetide. However, we may not exclude that the large difference in the occurrence of side effects between this study and available RCTs may be due to different study design (retrospective vs. prospective study) and/or methodology in evaluating side effects. In the present study, side effects were self-reported by patients to clinicians as commonly done in everyday practice, while in RCTs, the patients were instructed to complete an instrument, including a visual analog scale, assessing the presence and severity of nausea and a single question on whether the patient had vomited that day, each evening using an electronic device [3].

The occurrence of gastrointestinal disorders has clinical relevance because it frequently forces clinicians to administer medication at the sub-optimal dosage or in an intermittent way with either poor results and scarce adherence to therapy. In the literature, non-adherence to cinacalcet therapy varies from 46% to 71% because of gastrointestinal disorders [14].

## 5. Conclusions

The results of this observational study are of some clinical interests. They underscore the difficulty of targeting PTH levels with the available therapeutic strategies. They suggest that etelcalcetide represents a new valid therapeutic option for treating patients with SHPT. The treatment with etelcalcetide, in combination with other medications, may better control levels of PTH with low side effects. Besides, etelcalcetide may contribute to reducing the large pills burden that characterizes patients on dialysis, and, consequently, it may improve adherence to therapy, avoiding the risk of unremitting SHPT. Remarkably, etelcalcetide does not interact with cytochrome P450 and, for this reason, does not interfere with various drugs frequently used in dialysis patients [15]. It remains to be assessed whether etelcalcetide may improve the rates of parathyroidectomy, vascular calcification, or mortality. Being etelcalcetide very potent in suppressing PTH levels, even in severe SHPT, future studies should evaluate the potential risk of more adynamic bone disease during long-term therapy.

## Figures and Tables

**Figure 1 jcm-08-01066-f001:**
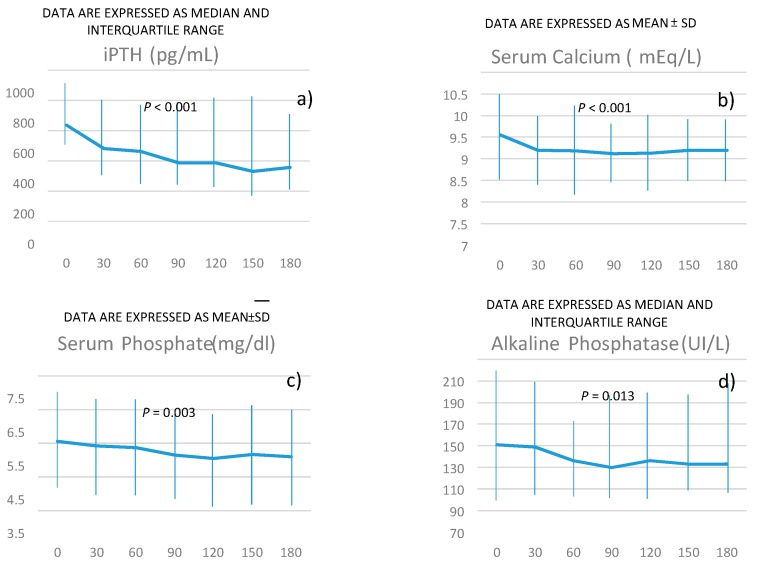
*P* for the trend of intact parathyroid hormone (iPTH) (panel **a**), serum calcium (panel **b**), serum phosphate (panel **c**), alkaline phosphatase (panel **d**) over time was obtained using linear regression models weighted for patients’ identification (see methods for more details).

**Figure 2 jcm-08-01066-f002:**
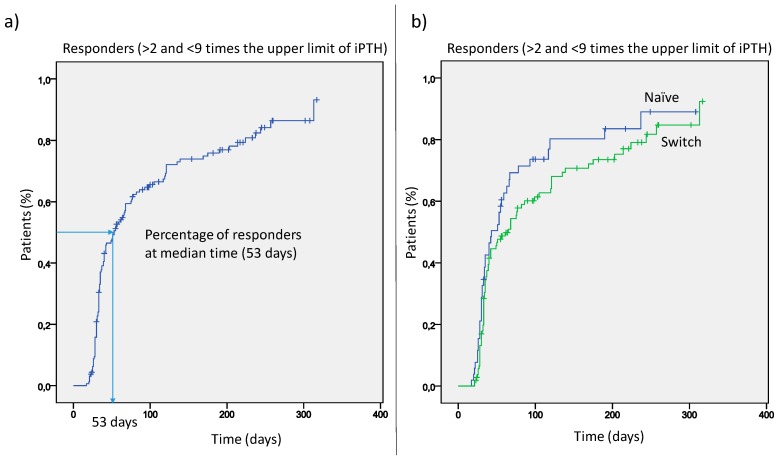
The median time for responders in the whole cohort (**a**) and differentiated for naïve patients and patients switched from cinacalcet to etelcalcetide (**b**).

**Table 1 jcm-08-01066-t001:** Patients’ Characteristics.

	Total Group (*n* = 168)	Naïve Group (*n* = 56)	Switch Group (*n* = 112)	*p* (Naïve vs. Switch)
Age (years)	61 ± 14	64 ± 14	59 ± 14	0.04
Male (%)	57	52	60	0.32
Dialysis vintage (month)	58 (IQR 32–102)	35 (IQR 14–63)	69 (IQR 48–120)	<0.001
Diabetes (%)	25	31	22	0.23
Cardiovascular comorbidities (%)	73	70	75	0.53
iPTH (pg/ml)	636 (IQR 493–916)	602 (IQR 509–800)	664 (IQR 495–947)	0.67
Serum Calcium (mEq/L)	9.0 ± 1.0	9.1 ± 0.7	9.0 ± 1.1	0.60
Serum Phosphate (mg/dL)	5.6 ± 1.4	5.5 ± 1.4	5.6 ± 1.4	0.83
Alkaline Phosphate (U.I./L)	131 (IQR 83–201)	111 (IQR 74–159)	148 (IQR 88–221)	0.02
Hb (gr/dL)	11.1 ± 1.4	11.0 ± 1.2	11.1 ± 1.4	0.58
ESA treatment (%)	87	88	87	0.97
Phosphate binders therapy (%)	96	93	97	0.17
Calcium containing binders (%)	18	25	14	0.09
Vitamin D therapy (%)	75	83	71	0.09
Native Vitamin D therapy (%)	5	4	6	0.59
Previous cinacalcet treatment (%)	67	0	100	N/A

IQR, interquartile range; iPTH, intact parathyroid hormone; Hb, hemoglobin; ESA, erythropoietin-stimulating agent.

**Table 2 jcm-08-01066-t002:** Cases of hypocalcemia.

Days after Parsabiv	<7.0 mEq/L	≥7.0 or <7.5 mEq/L	≥7.5 or <8.3 mEq/L
30	3/168 (1.8%)	0	25/168 (14.9%))
60	1/129 (0.8%)	7/129 (5.4%)	28/129 (21.7%)
90	2/111 (1.8%)	2/111 (1.8%)	27/111 (24.3%)
120	1/80 (1.3%)	1/80 (1.3%)	21/80 (26.2%)
150	1/61 (1.6%)	6/61 (9.8%)	11/61 (18.0%)
180	0	1/44 (2.3%)	11/44 (25.0%)
210	0	1/51 (2.0%)	15/51 (29.4%)

**Table 3 jcm-08-01066-t003:** Cause of discontinued treatment with etelcalcetide.

	30 Days (*n* = 129)	60 Days (*n* =112)	90 Days (*n* = 81)	120 Days (*n* = 62)	150 Days (*n* = 44)	180 Days (*n* = 51)
Hypocalcemia	4 (3%)	2 (2%)	1 (1%)	0	0	0
Gastrointestinal (G.I.) effects	1 (1%)	0	0	0	0	1 (2%)
Unavailability of the drug	1 (1%)	0	0	0	0	2 (4%)
Patients request	0	1 (1%)	0	0	0	0

**Table 4 jcm-08-01066-t004:** Adverse effects.

30 Days(*n* = 129)	60 Days(*n* = 112)	90 Days(*n* = 81)	120 Days(*n* = 62)	150 Days(*n* = 44)	180 Days(*n* = 51)
**4** **(3%)**	**Diarrhea**	**4** **(4%)**	**Stomach pain**	**3** **(4%)**	**Stomach pain**	0		0		0	
Stomach pain	Nausea(two patients)	Nausea(two patients)			
Nausea(two patients)	Unspecified

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
