# Peer review of "Etelcalcetide in Patients on Hemodialysis with Severe Secondary Hyperparathyroidism. Multicenter Study in “Real Life”"

_jcm, 2019, doi:10.3390/jcm8071066_

Round 1
Reviewer 1 Report
In this observational study included patients from hospital and university, a total of twenty-three dialysis units in different geographic areas of Itlay. Etelcalcetide is a new calcimimetic in intravenous form, and its’ efficacy has been ascertained in randomized controlled trials (RCTs). Authors evaluated in their study the safety of etelcalcetide in everyday clinical practice. However, the results seem not differ from RCT in clinical settings. Authors need to explain clearly why this study was done and population choo
In this observational study included patients from hospital and university, a total of twenty-three dialysis units in different geographic areas of Itlay. Etelcalcetide is a new calcimimetic in intravenous form, and its’ efficacy has been ascertained in randomized controlled trials (RCTs). Authors evaluated in their study the safety of etelcalcetide in everyday clinical practice. However, the results seem not differ from RCT in clinical settings. Authors need to explain clearly why this study was done and population choose. I also have major concerns regarding the measurements, analyses and conclusions of this paper.
Question 1.
Experimental Section
1. Title “study design” is more appropriate than “experimental design”.
2. The authors do not describe in that Etelcalcetide was administered to naïve patients (i.e. patients never treated before with calcimimetic) with SHPT and to patients who were on treatment with cinacalcet; in the latter condition etelcalcetide was administered as replacement medication (switch-group) either to obtain a better control of serum levels of PTH and to reduce side effects or pills burden. However, they need to explain how long, how much, what are side effects, how patterns of PTH levels changes, reasons to switch the medicine etc. in the switch group before switching the medicine. A figure of patient selection and criteria should be drawn to appropriately describe the study design.
3. The aim of the study and endpoints of the study (e.g. primary and secondary end points) should be mentioned properly.
4. The following levels of serum calcium were used for the definition of hypocalcaemia: < 7.0 mEq/l; ≥ 7.0 but ≤ 7.5 mEq/l; ≥ 7.5 but < 8.3 mEq. What the values represent and according to what scales?
5. “Clinicians were asked on gold range of PTH levels they attempted to obtain in clinical practice” Please describe clearly what it means.
6. Co-morbid conditions and drugs used should be added.
Result Section
1. Authors said that data were collected from twenty-three dialysis units with n.1190 patients on charge. Patients on treatment with etelcalcetide were n.161 (13,5%). Is that mean your analysis is on only 13.5% of patients who take etelcalcetide?
2. In table 1, authors need to describe clearly the data of naïve and switch groups to differentiate between the two groups. Which criteria used to switch the medicine? The number and percentage of patients are not well described in table 1.
3. In figure 2, authors describe that serum PTH declined over time from a median value of 636 pg/ml to 357 pg/ml. How the two groups (cinacalcet and etelcalcetide respond to treatment differently should be mentioned clearly.
4. Authors describe that median time for responders was 53 days and percentage of responders progressively increased during the further course of treatment and it was similar either in patients switched from cinacalcet and in naïve patients. Please describe clearly how the patients respond to cinacalcet before switch to etelcalcetide?
5. What is the figure legend “P FOR TREND OF iPTH, SERUM CALCIUM, SERUM PHOSPHATE, ALKALINE PHOSPHATE OVER TIME WAS OBTAINED USING LINEAR REGRESSION MODELS WEIGHTED FOR PATIENTS IDENTIFICATION” mean? Please describe more clearly.
6. Authors describe that all clinicians reported that “the preferred gold levels of PTH were between 80 and 500 pg/ml”. Which guideline levels of Ca, P, PTH used in your country? Please more clearly defined.
7. Table 3 legend. Cause of interruption of parsabiv. What does it mean?
8. Compare and describe the reasons for withdrawal from cinacalcet and etelcalcetide in the table 3.
9. There are many discrepancies in the study design and result section. e.g. Authors explained in result section as follows: In contrast to RCTs, no supplementation of calcium, increments of vitamin D sterols, or increase of calcium in dialysate were done in our cohort. And in methods section, authors describe as “On the basis of their own judgment and data analysis, clinicians modified the initial dose of etelcalcetide (5 mg three times per week, as suggested by manufacturing company); in addition, they adjusted phosphate binders, vitamin D sterols and supplement of calcium as needed.”
10. What is the following paragraph meaning? Is there any description error?
The results of this observational study are of some clinical interests. They underscore the difficulty of targeting PTH levels with the available therapeutic strategies. They suggest that etelcalcetide represents a new valid therapeutic option for treating patients with SHPT. The treatment with etelcalcetide in combination with other medications may better control levels of PTH with low side effects. In addition, etelcalcetide may contribute to reduce the large pills burden that characterizes patients on dialysis and consequently it may improve adherence to therapy avoiding the risk of unremitting SHPT. Remarkable, etelcalcetide does not interact with cytochrome P450 and for this reason does not interfere with various drugs generally used in dialysis patients.
Reviewer 2 Report
This is an interesting and timely study by renowned authors in the field about the effect of this new drug designed for control of secondary hyperparathyroidism in dialysis patients, showing that it is possible to improve adherence (and tolerance!!!!) of patients to previous administration of calcimimetics (cinacalcet) in a "real world" setting.
Minor suggestions:
1) Abstract: please add "in dialysis patients" in the first sentence.
2) Is there a particular reason to collect samples at the start of the FIRST dialysis of the week. Is it not more usual to collect them mid-week? You know it may affect absolute values of biochemical parameters and this aspect should be stressed elsewhere.
3) The definition of "responders" may be improved. It is not clear how the interval 80-500 was taken from beyond "all clinicians reported...". This is probabñy the only minorweakness of this interesting report that I enjoyed reading. Probably the results are similarly good by using more classical 2X-9X or 2X-5X or 150-300 pg/ML (KDIGO, KDOQI references).
4) It is especially interesting, as well as it is the current experience of many, that etelcalcetide is better tolerated that cinacalcet despite the absence of differences in the original RCT. How do authors explain that? Please, further discuss this important issue. How dosing of the physicians could be different from the RCT? (elapsed time to increasing does? How the prospective and retrospective nature of the study (and methodology to evaluate GI symptoms may have influenced? What it seems clear is that patients intolerant to cinacalcet DO tolerate etelcalcetide.
5) Please mention that the 8.3 mg/dl of calcium as chosen according to the Summary of Procudt Chracteristics
6) Could you please provide more information about Vit D use (native and active) as well as Ca bath 1.25-1.30.
7) it is clear that etelcalcetide is potent. Is there a potential risk of more adynamic bone disease than with cinacalcet? (bone studies probably will not be done as in BONAFIDE, but POTENTIALLY it should be another important end-point to mention
Congratulations
Some spellings and word should be reviewed in text, tables or figures (i.e. Figure 1 phosphate, STOMIC paton Table 4...))
Round 2
Reviewer 1 Report
Appropriate revisions make